# Danish Plastic Mass Flows Analysis

**Edward Vingwe [1],\***  **, Edgar Towa [2] and Arne Remmen [1]**

[1] Department of Planning and Development, The Technical Faculty of IT and Design, Aalborg University, 9000 Aalborg, Denmark; ar@plan.aau.dk

[2] Institute for Environmental Management and Land-Use Planning, Université Libre de Bruxelles (ULB), 1050 Brussels, Belgium; etowakou@ulb.ac.be

\* Correspondence: evingwe@plan.aau.dk

**Abstract:** In this paper, material flows and resource potentials for plastics at a national level in Denmark are mapped using an Environmentally Extended Multiregional Input-Output (EE-MRIO) database. EE-MRIO offers an operative improvement to current and prevalent methods for assessing the industrial and societal metabolism of resources, including plastics. The Exiobase is applied to map (1) the major sources, (2) calculate the total supply, (3) uses of plastics and waste generation, and (4) end of life pathways in order to indicate the potentials of plastics in the circular economy in Denmark with a focus on recycling. Furthermore, it elaborates how and why this method for performing Mass Flow Analysis (MFA) differs from mainstream assessments of material flows and from default uses of national statistical data. Overall, the results are that Denmark has a total supply of ≈551 kilotonnes (Kt) of plastics, out of which ≈522 Kt are used domestically and ≈168 Kt of plastic waste are generated annually. Out of the yearly amount of plastic waste, ≈50% is incinerated and 26% is recycled. These results indicate significant potentials for applying circular economy strategies and identify relevant sectors for closing the plastic loops. However, other initiatives are necessary, such as improvements in product design strategies, in the collection and sorting systems as well as in cross-sectoral collaboration.

**Keywords:** circular economy; sustainability indicators; sustainable consumption and production; mass flow analysis

## 1. Introduction

Plastics have gained societal attention both at international, EU, and national levels. The primary attention relates to the fact that the vast majority of plastics are produced from fossil hydrocarbons [1]. Additionally, sustainability issues associated with plastic evolve around its ecotoxicology and exponential growth in volume and velocity at which plastics are used and disposed of [1].

With a continuous increase in production, coupled with its low degradability and hazardous additives, plastic pollution and the associated high risks it poses to human health and marine, freshwater, and terrestrial environments have become global concerns [2–4]. In recent years, plastics pollution has gained public attention, because of the physical hazard to wildlife that ingests it or is entangled or lacerated by it [5]. Additionally, plastics disintegrate into microplastics, which can travel more easily from one environmental context to another [6].

However, due to their suitability in ubiquitous societal applications [7], the total amount of fossil oil-based plastics produced globally accumulated to 7800 Mt between 1950 and 2015 [1]; and 50% of this was produced during the last 13 years (ibid). Compared to bio-based and bio-degradable plastics whose global production capacity was only 4 Mt by 2017, the global production of fossil oil-based plastics increased with a compound annual growth rate of 8.4% from 2 Mt in 1950 and 380 Mt in 2015 (ibid).

In terms of plastic waste from a global perspective, Geyer et al. (2017) [1] reported that waste from primary and recycled plastics accumulated to 6300 Mt during the same period. Of this amount, ≈12% had been incinerated, ≈9% recycled and ≈60% were discarded either in landfills or natural environments [1]. Less than 10% of the ≈ 30 million tonnes of plastics found in municipal solid waste in the United States are recycled. Of the 26 million tonnes of post-consumer plastic waste generated in Europe in 2014, 30% was recycled and 40% incinerated [8]. The current situation in the European Union is that EU member states collect ≈ 27 Mt out of which, 31% (8.4 Mt) is recycled [8] and 63% (5.3 Mt) of the recycling takes place in the EU whereas, 37% is exported for recycling [9].

A main global plastic recycler, China, has banned the import of poor-quality plastics waste. China has imported and recycled a cumulative 111 Mt of plastic waste since 1992, 70% of which arrived from higher-income countries [10]. Additionally, the most recent UN amendment of the 1989 Basel Convention includes plastics waste in the same legally binding framework as other hazardous waste products [11,12]. To be implemented by 2020, the consequence of the amendment is that 180 governments have agreed to regulate the export of low-grade plastics waste that cannot be readily recycled to low-income member states. Any such exports will require the consent of the authorities of the receiving country [11].

To deal with the global increasing environmental awareness and pressures, societies around the globe are looking to find and implement sustainability strategies to effectively manage the use and end-of-life pathways of plastics. In 2018, the EU identified plastics as a key resource in its transition towards a circular economy that aims to increase the recycling of plastic packaging by 55% by 2030 [13]. Later the same year, the European Commission adopted the European Strategy for Plastics in a circular economy [14], which was followed by the Single-Use Plastics Directive in 2019 [15]. China's ban on low-grade plastics waste and the UN amendment of the Basel Convention, together with increased global awareness of the problems with plastic waste as well as the low recycling rates, are indicators that changes are underway.

The aim of this paper is to trace plastic flows in the supply chain via a Mass Flow Analysis (MFA) that is performed at a national level in Denmark. One specific objective is to highlight the potentials for the Danish plastics industry in a circular economy context. This was the focus in "Recycling of plastic waste—a strengthened Danish industry" that initiated this MFA in 2017 and is reported in Danish [16,17]. The motivation was to explore strategies to improve local recycling as estimates from the Danish Ministry of Environment and Food indicated that every 1000 tonnes of locally recycled plastics have the potential to generate 3–4 local jobs and at least DKK 6 million turnover [18]. The project began in 2017, in collaboration with the Danish Plastics Federation, with 37 firms to investigate the potentials for improving the amount and quality of plastic waste recycling. The research also focused on identifying relevant actors for the purpose of initiating collaboration.

Exiobase has been used to perform this MFA, however, note that the Exiobase contains monetary and physical supply and uses tables with data from 1995 up to 2011. During the scoping of this research project, there was a need to gain knowledge about the current situation, therefore, the most recent trade index data were used to nowcast the results of this MFA.

Exiobase is a scientific method for performing MFAs which differs from most mainstream methods. MFAs are becoming instrumental tools to assist policymakers to make strategies and policies in their transitions towards the Circular economy (CE) strategies [19]. However, the majority of the MFAs performed insofar, have solely depended on statistical data (see for example, [20–22]. Inconsistencies such as inefficiency in waste separation and registration, as well as mismatches between the EU and Danish waste recovery and disposal codes, have been identified [20]. According to Takou et al., 2019 [20], the latter increases uncertainties of the MFA results based on statistical data.

Other methods include measuring plastic waste composition in solid waste based on biogenic carbon in combustible waste [23]. This is the source that has been used to estimate fossil carbon in waste for incineration in Denmark's National greenhouse gas (GHG) inventory. According to this source, the content of fossil carbon in household waste is 79 kg/tonne wet mixed waste, and plastic

contains 0.672 kg C/kg [23]. By calculation, this method estimates that Denmark incinerates 350 Kt., an estimate which can be argued to be at the high end. Considering that the total supply, stock additions, and lifetime of plastic products in various Danish sectors, the differences between these various methods indicate that there are significant uncertainties about the amount of plastic waste in Denmark. There is, therefore, a need for a standard method for performing MFAs, most especially for the CE transition [20]. This paper demonstrates and elaborates on how and why Exiobase could be such a standard method for effective CE strategies (see Section 2).

## 2. Materials and Methods

The research uses multiregional hybrid-units input-output database Exiobase v3.3 to map the sources, total supply, uses, waste generation, and end of life (EoL) pathways of plastics in Denmark. Exiobase (http://www.exiobase.eu/) is a product developed by a European research consortia through four large scale EU projects (FORWAST (), EXIOPOL (), CREEA () and DESIRE ()) [24]. This assessment method is backed by two decades of work on integrating environmental and economic accounting [25–30]. Refer to [24] for information about how the Exiobase is constructed and how the database can be applied (also see [19]).

The database is based on environmentally extended multi-regional supply-use tables (EE MRSUTs) and input-output tables where physical supply and use tables (PSUTs) are provided. Based on economic activities and environmental data across countries, EE MRSUTs data provides consistent in-depth scientific insights in relation to how products are produced and used, and where resources are extracted, waste is generated and emissions are discharged [19]. See [24] for a concise description of the methodology i.e., the construction of PSUTs of Exiobase v.3. Data in PSUTs is becoming increasingly essential as the key to unlocking useful information required as a knowledge basis for designing effective policy strategies to decouple the socio-economic metabolism from its current level of environmental impact [19].

The latest hybrid version Exiobase v.3.3 presents a total overview of inputs and outputs from 164 industrial sectors in 43 countries and 5 'rest-of-the-world' regions [24]. These inputs and outputs include products in physical units and services in monetary units and (tonnes and kWh), emissions, natural resources, and wastes.

Exiobase v. 3.3 has been chosen specifically because:

- It provides PSUT, which is the basis of the MFA.
- It provides consistent monetary and physical SUTs, enabling the coupling of economic data with physical MFA data and energy data.
- Industries and products are distinguished at a high level of granularity (164 sectors and 200 products).
- It represents a closed mass balance at several levels: Total supply and use of products, total inputs and outputs of industries, as well as the fate of every input (product, resource, or waste), are ensured to be in balance with the corresponding outputs of products, emissions, wastes, and stock additions. The latter involves the calculation of waste flows [24,31,32].
- It distinguishes the production of virgin materials from the recycling of wastes (from which new materials are produced).

Further, all products in the Exiobase are organized in a standardized supply-use framework, and the individual countries' accounts are linked via world trade statistics. Thereby, the model consists of mass flow data for all products and waste fractions in all countries. In addition, this database also enables the life cycle assessment calculations of different uses of waste, e.g., recycling, incineration, landfilling, composting, and biogas for a wide spectrum of waste fractions.

Furthermore, not only does EE MRSUT provide in-depth data for understanding the complex net of global economic relationships and their environmental consequences but also presents economic and environmental data in a format that is consistent with the recommended accounting systems

proposed by the United Nations (UN) System of Environmental-Economic Accounting (SEEA) [33]. Another distinctive factor in the Exiobase is related to the waste calculation approach. There are two approaches for calculating waste i.e., the Life Cycle Assessment (LCA) or Exiobase.

The LCA assumes a convoluted time approach whereby the total supply of products is equivalent to total waste with immediate effect from when the product enters the use phase. This approach assumes that all resource extractions, manufacturing processes, use activities as well as the disposal of products, all occur simultaneously.

On the contrary, the Exiobase assumes a time series approach through which product lifetimes and stock depreciation, as well as stock addition, are considered in the calculation. One current example that demonstrates this approach of waste calculation relates to the global plastics MFAs [1]. This study shows how various product lifetimes of plastic used in eight different sectors (i.e., packaging, consumer and institutional products, other and textiles, electrical and electronic, transportation, industrial machinery, and building and construction), play a significant role in the rate at which plastic waste will be generated.

Furthermore, waste accounts in the Exiobase are based on the definition that waste is "materials for treatment" i.e., materials that need to be treated before being reintroduced in the economy or disposed of in the environment [24,32]. This definition assures that the law of the conservation of mass is respected in all national economic activities.

Waste accounts derive from a general procedure determining Physical Supply and Use Tables (PSUTs) or Hybrid Supply and Use Tables (HSUTs) which include a physical layer developed through three EU projects: (1) FP6 project FORWAST [31], (2) FP7 project CREEA [32,34] and FP7 project DESIRE [19,24].

The term 'waste' in the Exiobase denotes a broader meaning than that commonly used because it includes both waste residuals and waste products [32,35]. Waste accounts are thus, divided into two sets [32]. The first relates to the users and the other, to the producers of waste. Waste treatment activities are essentially filling the first account, whilst the activities and final consumers encompass the second. In practice, a third account is produced showing unregistered waste, which equals the supply of waste less than the overall use of waste. This involves waste that by-passes the industrial waste treatment processes, e.g., illegally disposed of waste.

The basic waste accounting approach in the Exiobase firstly calculates the supply of waste based on a mass balance approach, where all inputs are subtracted by the supply of products and emissions equals the supply of waste and stock additions. For example, if a packaging manufacturer uses 1 tonne of plastics and produces 0.98 tonnes of packaging, then the supply of waste plus stock additions is 0.02 tonnes. This waste calculation is carried out per homogenous fraction of materials, however, the waste calculation in the Exiobase is more sophisticated than this, see e.g., [31,32]. In Exiobase v.3.3 however, the waste plus stock additions are determined by the exogenously specified waste treatment data plus trade with waste, which are collected from waste statistics. Hence, the overall supply of waste in the Exiobase waste accounts is determined by available data in waste statistics.

Since the data quality and coverage of national waste statistics are relatively low, waste calculations in the Exiobase are likely to be affected by discrepancies (that have not been estimated) inherent to data collection and organization conducted by the national statistical agencies. However, given the physical supply and table of use in the Exiobase, it is possible to compare the waste amounts with the use of products, which theoretically should be the same in the long term in a steady-state economy. Obviously, the real economy is not in a steady-state and so in the long run, this comparison does not always provide a good estimate of the error in waste accounts. However, at least it gives an indication of the error.

The part of Exiobase that concerns Denmark is based on average Danish industries and product categories. The economy is divided into 164 sectors which trade with each other within 200 different product categories. This level of relatively highly aggregated sectoral and product classification gives rise to a number of limitations when identifying specific products and their use. At EoL for example,

plastics used in multi-material products, such as electronics, furniture, and construction are respectively registered either as or electrical and electronic equipment waste (WEEE) or waste from furniture or construction. Therefore, it is not accounted for as plastic waste in national waste accounts.

Additionally, due to the high sectoral detail in the Exiobase and the instantaneous change of product/sectoral classification from NACE 1 to 2 may require nation-specific correspondence and visual checks for concordance. Related to this case, the Exiobase indicated, for example, that there do exist an industry sector that produces plastics, basic. However, per definition, the actual production of plastic, basic involves the transformation of oil and gas into primary forms of plastics. Per definition, this sector does not exist in Denmark. Therefore, domestic production of plastic, basic as indicated in the Exiobase, may imply the Danish plastics industrial activities specialized in value-adding processing of primary forms of plastics. Such processes as pigmentation and productions of special plastic products such as foils, slates, etc.

Overall, the Exiobase EE MRIO database provides a consistent framework for tracking emissions, resource use, and other environmental pressures along global supply chains thereby linking consumption patterns to production processes elsewhere [19]. According to Stadler, et al. (2018), the high sectoral detail and wide spectrum of environmental data allow for both economy-wide assessments as well as the identification of environmental hotspots. Exiobase V3 provides researchers and policymakers with a unique tool to identify priority sectors and consumption areas. Thereby giving them the ability to assess priorities of policies set in place to reduce environmental impacts, increase resource efficiency, and ultimately, to decouple the economic activity from the environmental impacts [19].

## 2.1. The Scope of the MFA

The scope of this MFA is to map the sources, total supply (both from imports and domestically processed plastics), uses, waste generation, and EoL pathways of plastics in Denmark. Note that excluding trade with plastic including waste (export), plastic waste treatment practices in Denmark are recycling at the material level, quaternary recycling (incineration for energy recovery), and landfilling.

Further, the total supply herein represents the total input i.e., the sum of imports and domestic productions of plastic in Denmark. The difference between total inputs, uses, waste recovery, and treatment and exports represents stock addition and/or plastic used in multi-material products or long-life products and illegal trade with plastic waste. As one of the boundaries of this MFA, illegal trade with plastic waste remains undocumented as it is scientifically unjustifiable, and therefore excluded. Further, incinerated plastic waste in Denmark is for quaternary recycling, however, the amount of energy recovered from plastic waste has been excluded. This has been delimited since energy recovery is disregarded as a CE strategy in the EU [36] (p. 228) and [37] (p. 292). However, the MFA shows the quantity of incinerated plastic to indicate the lost potential for recycling. Furthermore, the Exiobase contains different types of emissions data, however, emissions from plastic will not be presented since this information is out of scope.

## 2.2. The Multi-Regional Hybrid Supply and Use Tables (MR-HSUTs) Data Sources

Data used from the Exiobase include the total supply and use of plastics, total inputs and outputs of industries, and final consumption (households), as well as the fate of every input (product, resource, or waste). This is ensured to be in balance with the corresponding outputs of products, emissions, wastes, and stock additions. The latter involves the calculation of waste flows [24,31,32].

In terms of the supply, the Exiobase contains monetary and physical supply tables with data for 200 products (in tonnes (t), Euro (EUR), and in tele joules (TJ) for energy i.e., electricity and heat). Altogether, these products constitute all types of products on the market and, plastic is one of them. All products are organized in unit EUR and physical products, in tonnes. The monetary approach corresponds with the total turnover per industry and there is data on domestic production and import of plastic for Denmark.

In terms of uses of the 200 products, Exiobase includes the same data as access (supply) of products. Thus, there is data on the uses of plastics in Denmark which is disaggregated into 164 industries and final consumption (households).

Exiobase also contains data for 17 different homogenous waste fractions, including plastic (in tonnes). In terms of EoL pathways (waste treatment), there is data available for the amounts (in tonnes) of these fractions that go for treatment (recycling, incineration and landfilling, etc.). This MFA used this data to calculate the quantity of plastic waste generated in Denmark and made it possible to map both the amount and treatment of plastic waste.

Additionally, the Exiobase contains data of different types of emissions and resources (in tonnes). To be specific the database contains data for 50 emissions to air, 3 emissions to water, 9 emissions to soil, and 34 different resources. However, emissions from plastic production, consumption, and treatment activities are out of the scope of this MFA.

### 2.3. Calculation Procedure

- Production of plastics, basic: Data on production is obtained as an elaboration of IEA data (Stadler et al., 2018), which provides information on the energy products converted into plastics. Inputs of feedstock materials are converted into plastics by transfer coefficients [38]. Note that plastics, basic refers to plastics and synthetic rubber in primary forms (Eurostat, 1996), whereas domestic productions herein refers to productions of plastics taking place in Denmark. Further, the analysis traces and disaggregates the import and export flows of plastics per importing and exporting country, source, and quantity.

- Production of rubber/plastic products: Data obtained endogenously by the Exiobase algorithm [24]. In practice, the production and imports of plastics and other feedstock materials are distributed according to Exiobase monetary supply and use tables (MSUTs) [19]. These inputs are then multiplied by transfer coefficients [38] in order to obtain the total supply.

- Data on trade of plastics, basic and rubber/plastic products: These values are obtained as an endogenous value from the Exiobase algorithm [24]. The estimation is based on the trade bilateral flows calculated in Exiobase in monetary units [19].

- Use of plastics, basic and rubber/plastic products: These amounts are obtained by the Exiobase algorithm. The total availability of plastics in the Danish economy is constrained by total supply and trade flows. The distribution of plastics to the economic agents is based on what is indicated in MSUTs considered as a constraint in respect to mass balance.

- Recycling of plastic waste: Data from Eurostat [39] is used for the collection of plastics waste. Then data from COMTRADE [40] is then used for the trade of waste. Collection of plastics waste minus export plus import gives the number of recycled plastics.

- Incineration and landfilling of plastic waste: Initial data on the collection and treatment of material has been collected from Eurostat for the years 2006 and 2008 [39]. When Eurostat waste data was not disaggregated into waste fractions but accounted as municipal waste, we disaggregated it using data from Christensen (1998) [41].

### 2.4. Data from Statistics

Although this MFA uses the latest version of Exiobase, the database is compiled based on a time series of EE MRIO tables ranging from 1995 to 2011 [19]. However, during the scoping of this MFA, there was, a need to gain knowledge about the current situation. Therefore, statistical data from sources other than the national statistical offices were used to compile waste accounts in Exiobase and has been used to nowcast this MFA. The most recent trade index data has been used for now-casting. Thus, laying the basis for the now-casting (Section 4), the first section (Section 3) of the MFA results is based on the Exiobase data from 2011. Some flows were adjusted to meet a mass balance criterion and others, by using the same distribution as in 2011.

When using statistical data, the percental change from 2011 to 2017 has been identified, and this is then applied to the data for 2011 to calculate the flow for 2017. An example of the use of a mass balance criterion is when we have adjusted data on import, export, and production. Here the use has been calculated as the sum of production and import minus export. An example of the use of the distribution of flows from 2011 is for the sectoral use as well as waste generation, where the same sectoral distribution of the total use and the total waste as for 2011 has been applied for 2017.

Below, we outline and indicate how the different data on flows for 2011 was adopted and used to now-cast the flows for 2017 economic activities:

- *Import of:*

  - Plastics, basic: 30% decrease from 2011–2017 based on [40];
  - Rubber and plastic products: 27% decrease from 2011–2017 based on [40].

- *Export of:*

  - Plastics, basic: 18% increase from 2011–2017 based on [40];
  - Rubber and plastic products: 21% decrease from 2011–2017 based on [40]

- *Domestic production of:*

  - Plastics, basic: Constant from 2011–2016 * based on [8,42–46] and
  - Rubber and plastic products: Constant from 2011–2016 * based on [8,42–46]

- *Waste treatment and recycling of plastics waste:*

  - Recycling: 5% increase from 2010–2016 ** [47];
  - Incineration: 15% increase from 2010–2016 ** [47]
  - Landfill: 57% decrease from 2010–2016 ** [40]

  * Data for 2016 are used to represent 2017
  ** Data for 2010 and 2016 are used to represent 2011 and 2017 respectively

  Data calculated based on mass balance criteria:

- *Domestic production of:*

  - Secondary polymers (recycling): Calculated as the recovery of plastics waste in Denmark (see above) multiplied by a loss at 10%

- *Domestic use of:*

  - Plastics, basic = domestic production + import − export
  - Rubber and plastic products = domestic production + import − export

- *Non-specified output* = domestic use (see above) − recycling − incineration − landfill (see above)

  Data where the same distribution as in 2011 has been applied for 2017

- Distribution of total use of plastics on sectors and households.

## 3. Results

In this section, we present and describe our results in five flows i.e., (1) the major plastic sources, (2) total supply per source, (3) total uses per consuming activity, and (4) total waste generated including (5) their end-of-life pathways (treatment). For the purposes of this project, other consuming activities or sectors than agriculture, food industry, chemical (including pharmaceuticals), retail, construction, plastics industry, and households have been categorized as 'other'. See Figure 1 for an overview (the thicker the flow the more significant it is by quantity.

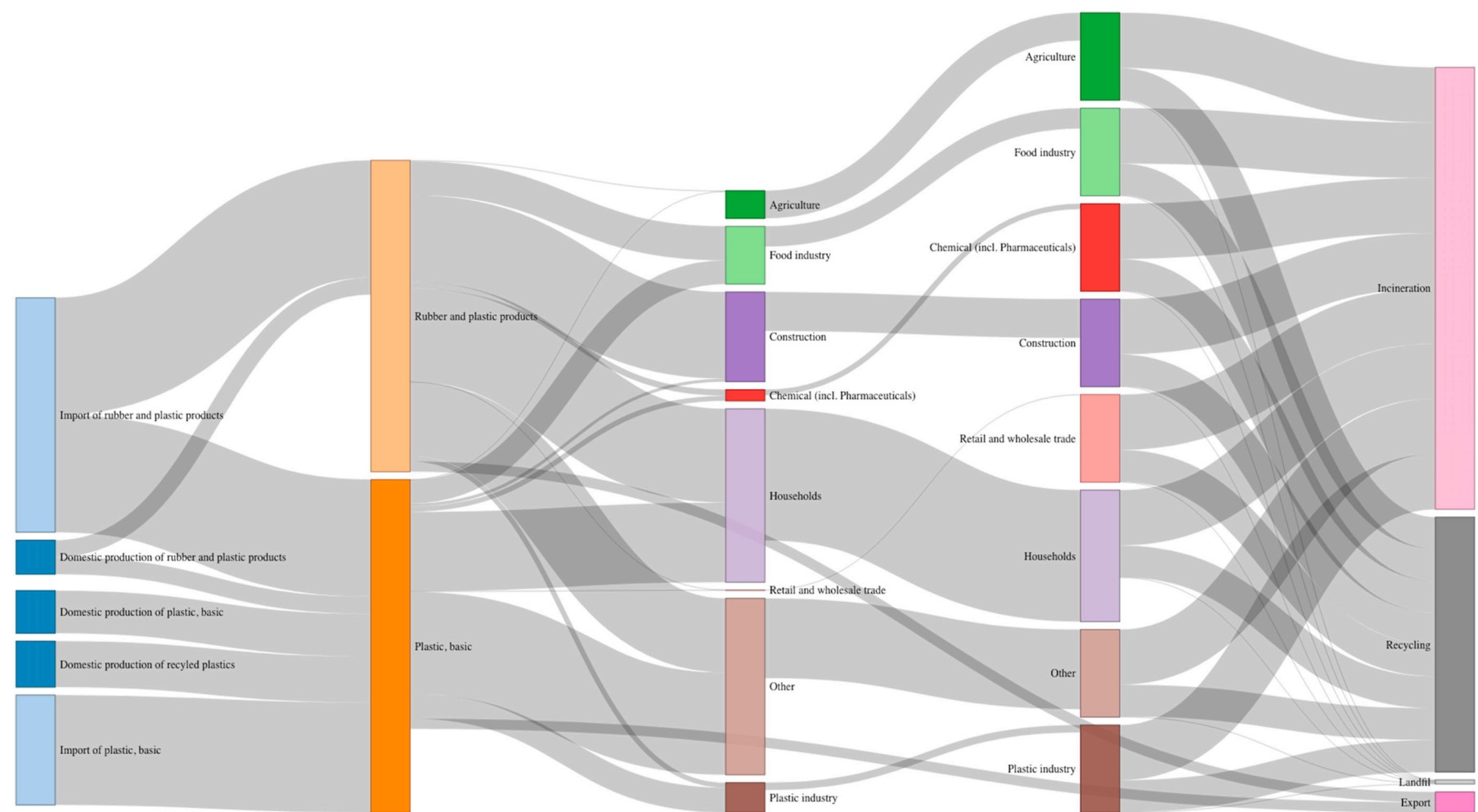

**Figure 1.** An overview of plastics flows in Denmark. Derived after the 2011 Exiobase database. The Sankey diagram has been created using RStudio and the flows are scaled in proportion to the amounts (quantity) per flow.

Quantities and descriptions of each flow are provided in the following Section 3.1 (total supply), Section 3.4 (total uses), and Section 3.7 (waste). Moreover, the energy recovered from incineration is also not shown as it is out of the scope of this study. The European Union has excluded incineration for energy recovery as part of its circular economy strategy [36] (p. 228) and [37], (p. 292).

Further, in reference to Figure 1, the three major sources of plastics in Denmark are in general, (1) plastics, basic, (2) rubber and plastics products, and (3) secondary polymers from recycling. Overall, domestic productions of plastics in Denmark represented ≈26% (144 Kt) of the total supply, whereas ≈74% (404 Kt) was imported.

Further, the total supply of resources in Exiobase is disaggregated by quantity per source and origin (i.e., domestic and import supplies), as are the uses and waste generation. These are also key elements that show the trade balance of the Danish plastics industry. Conveniently, serving this MFA's main purpose, the total uses and the generated waste are disaggregated by quantity per source and per eight sectoral categories as introduced earlier on. With the exception of the activity category 'Other', the seven sectors have been categorized in accordance with the NACE product/sectoral classification in correspondence to the Danish industrial branch codes.

### 3.1. The Total Supply of Plastics in Denmark

Our MFA showed that the total supply of plastics in Denmark is ≈548 Kilotonnes (Kt) (excl. the import of ≈67 Kt plastics waste). The import of plastics, in general, is more significant than the domestic supply.

### 3.2. Domestic Supply

Of the total (144 Kt) domestic productions (in Figure 2), ≈35% (50 Kt) was plastics, basic, ≈38% (54 Kt) was secondary polymers, whereas ≈28% (40 Kt) was rubber and plastic products. Thus, the output from recycling constituted ≈10% of the total supply.

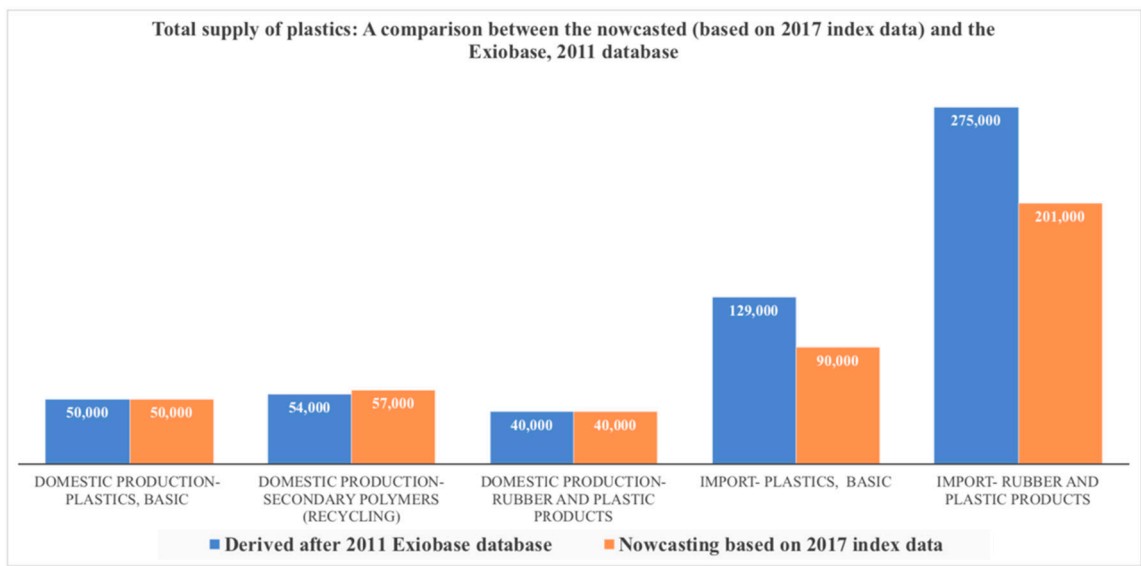

**Figure 2.** Total supply of plastics disaggregated in tonnes per source (excl. import of plastics waste): A comparison between the nowcast (based on 2017 index data) and the Exiobase, 2011 database.

Furthermore, it is noteworthy that due to the abrupt change of product classification from NACE 1 to 2, some discrepancies in terms of product/sectoral classification were observed. In this case for instance, although the Exiobase indicated that there do exist production of plastics, basics in Denmark, this may not be by definition, true. The actual production of plastics, basic i.e., the transformation of oil and gas into primary forms of plastics, basic do not exist in Denmark. Therefore, the domestic

production of plastics, basic as indicated in the Exiobase may have implied, for instance, activities related to the Danish plastics industry which perform specialized value-adding processing of primary forms of plastics such as pigmentation, and special plastic products such as foils and slates, etc. Hence, visual checks such as sectoral concordance/correspondence were performed to rectify this discrepancy proved necessary and recommendable.

*3.3. Import*

Import accounts for 74% (404 Kt) of the total supply of plastics in Denmark (in Figure 2). Of the total imports, plastics, basic accounts for ≈32% (129 Kt) whereas, rubber and plastic products account for ≈68% (275 Kt). With 1 Kt cut-off, 16 significant countries exporting plastics, basic to Denmark were identified. Out of which, the top four most significant exporting countries (exporting ≥ 10 Kt) were the Netherlands (≈35 Kt), Germany (≈22 Kt), Belgium (≈18 Kt), and Norway (≈10 Kt).

Furthermore, in relation to the 275 Kt of imported rubber and plastic products into Denmark, we identified 26 of the most significant exporting countries (exporting ≥ 1 Kt). Out of which the top six most significant exporting countries (exporting ≥ 10 Kt), were the Netherlands (≈59 Kt), Rest of World Asia and Pacific (58 Kt), Germany (38 Kt), Sweden (16 Kt), China (15 Kt from), and Czech Republic (11 Kt). Further, in terms of trade with waste, Denmark imported ≈67 Kt of plastics waste.

*3.4. Total Uses of Plastics in Denmark*

Of the total supply of plastics, our MFA showed that Denmark used a total of 551 Kt tonnes. The uses by quantity per source, per domestic and export uses have been disaggregated.

*3.5. Domestic Use*

Overall, the total domestic uses of plastics in Denmark precedes that of exports. Domestic use herein refers to the sum of all national plastic consuming activities excluding exports. The significant domestic uses of plastics by quantity per source in Denmark is that of rubber and plastic products followed by that of plastic, basic. Of the total uses, rubber and plastic products account for ≈53% (291 Kt including internal use) whereas plastic, basic accounts for ≈42% (231 Kt).

Furthermore, we disaggregated the uses in tonnes per source per consuming activity (sector) in order to gain an overview of the most significant plastic consuming activities in Denmark. For the scope of this project, consuming activities or sectors that do not fall under agriculture, food industry, chemical (including pharmaceuticals), retail, construction, plastics industry, and households have been categorized as 'other'.

Of the total domestic uses of plastic, basic, the results are that households use ≈33% (77 Kt), plastic industry ≈10% (24 Kt), food industry ≈10% (23 Kt), chemical including pharmaceuticals ≈2% (5 Kt), and construction 1% (3 Kt), whereas the retail and agricultural sector used ≈0.2% (400 t) and 0.1% (200 t). The remaining ≈43% (99 Kt) were used in other sectors which overall, constituted 10 among the 14 topmost significant consuming sectors of plastic, basic (using ≥ 3 Kt) in Denmark. See Figure A1 in Appendix A showing in specific, these sectors and their quantities of use. Collectively, these 10 sectors account for ≈22% (52 Kt) of the total uses of plastic, basic in Denmark.

Of the total domestic uses of rubber and plastic products in Denmark, then households account for ≈31% (91 Kt), construction ≈29% (84 Kt), food industry ≈11% (33 Kt), chemical including pharmaceuticals ≈2% (6 Kt), plastic industry 2% (3 Kt), the agricultural sector ≈0.2% (700 t), whereas the retail sector used none. The remaining ≈25% (72 Kt) were used in other sectors which overall, constituted 6 among the 12 topmost significant consuming sectors of rubber and plastic products (using ≥ 3 Kt) in Denmark. See Figure A2 in Appendix A showing in specific, these sectors and their quantities of use. Collectively, these 6 sectors accounted for ≈18% (45 Kt) of the total uses of this source of plastics.

*3.6. The Danish Export of Plastics*

Excluding the trade of waste, Denmark exported the sum of ≈29 Kt of plastics, out of which ≈92% (27 Kt) was rubber and plastic products and the remaining ≈ 8% was plastics, basic. Sweden (≈483 t), Germany (≈268 t), Poland (≈145 t), Japan (≈123 t), and Great Britain (≈120 t) were the most significant importers of plastics, basic from Denmark. At a 10 tonne cut-off, however, we identified 31 countries importing plastics, basic from Denmark.

In relation to rubber and plastic products and exporting ≥ 1 Kt, Germany (≈5.238 t), Sweden (≈3.274 t), Norway (≈2.686 t), Great Britain (≈1.589 t), and France (≈1.506 t) were the most significant importers of rubber and plastic products from Denmark. At a 10 tonne cut-off, we identified 34 countries importing this source of plastics from Denmark. Furthermore, in terms of the trade with waste, Denmark exported ≈ 51 Kt of plastics waste.

*3.7. Total Supply of Plastics Waste in Denmark*

This sub-section is an analysis and presentation of the amount of plastics waste domestically generated in Denmark. The analysis also includes the end-of-life pathways of plastics waste in Denmark. The MFA results showed that ≈168 Kt of plastics waste were produced from all the Danish economic activities. Thus, including the 67 Kt from imports, the total supply of plastics waste in Denmark amounts to ≈235 Kt.

Domestically generated plastics waste per eight activities/sectors amounted to 168 Kt in tonnes. Setting aside category 'other', the most significant activities/sectors by the total mass of plastics waste produced were households producing ≈42% (71 Kt), construction ≈13% (21 Kt), agriculture ≈9% (15 Kt), and the food industry ≈7% (11 Kt), whereas the plastics industry, chemical (including pharmaceuticals), and retail respectively accounted for 2% (4 Kt), 2% (3 Kt) and 0.2% (90 t). The remaining ≈26% (43 Kt) were produced by other sectors which overall, at a 500-tonne cut-off, we learned that ≈73% (22 sectors) were among the top 30 most significant producers of plastics waste in Denmark. Collectively, these 22 sectors accounted for ≈20% (33 Kt) of the total plastics waste generated.

*3.8. End-of-Life Treatment Pathways of Plastics Waste in Denmark*

Our MFA showed that of the ≈235 Kt total supply of plastics waste in Denmark, ≈44% (104 Kt) were incinerated for energy recovery, ≈26% (60 Kt input) were recycled in Denmark i.e., giving an output of 54 Kt recycled plastics, 22% (51 Kt) were exported, 0.4% (1 Kt) were landfilled, whereas the EoL of 8% (19 Kt) remains unspecified. Note that the 60 Kt (input) for recycling is a calculated quotient of 54 Kt tonnes (output) and 0.90 based on recycling efficiency. The latter average standard of 10% materials loss is based on the Danish plastic recycling processing industry.

## 4. Now-Casting of the Supply, Use, Waste Generation and EoL Treatment of Plastics in Denmark (Based on 2017 Index Data)

The previous section mapped the plastic flows in Denmark based on the Exiobase database, 2011. These results laid the basis for this section in which, we use the 2017 data index (presented in Section 2) to nowcast the current situation. Said differently, the data for 2017 are largely based on the data for 2011, but in this section, we use the most recent index data to adjust the main flows to reflect 2017.

Comparing 2011 with nowcast flows, our results in Figure 3 below show that the total supply of plastics in Denmark decreased by ≈20% i.e., from ≈548 Kt (2011) to ≈438 Kt (2017) (excl. the ≈37 Kt import of plastics waste).

Notable also, is the ≈30% decrease in imports of plastics, basic and ≈27% that of rubber and plastic products. While domestic productions of (1) plastics, basic, (2) rubber, and plastic products have both remained steady, the output from domestic recycling increased by ≈5%.

Further, the total uses of plastics in Denmark decreased by ≈21% i.e., from 551 Kt (2011) to 438 Kt (2017). See Figure 3 below.

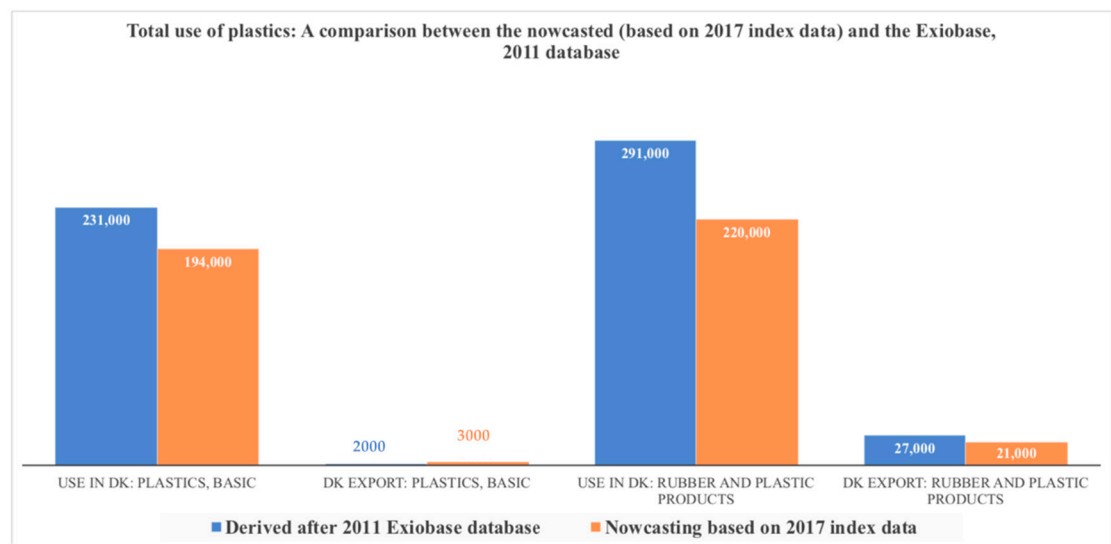

**Figure 3.** Total use of plastics disaggregated in tonnes per source: A comparison between the nowcast (based on 2017 index data) and the Exiobase, 2011 database.

Notable also, is the ≈15% decrease in the uses of plastics, basic and ≈24% that of rubber and plastic products, as is the ≈22% (from 27 Kt to 21 Kt) decrease in the Danish exports of rubber and plastic products. However, the Danish export of plastics, basic increased by ≈33% (from 2 Kt to 3 Kt).

Furthermore, in relation to the total supply of plastics waste in Denmark, overall, we observed a ≈2% increase, from 235 Kt by 2011 to 239 Kt by 2017. The Danish import of plastics waste decreased by ≈55% (from 67 Kt to 37 Kt). However, its export and incineration activities respectively, increased by ≈9% (from 51 Kt to 56 Kt and 13%. See Figure 4 below comparing EoL pathways for 2011 to 2017.

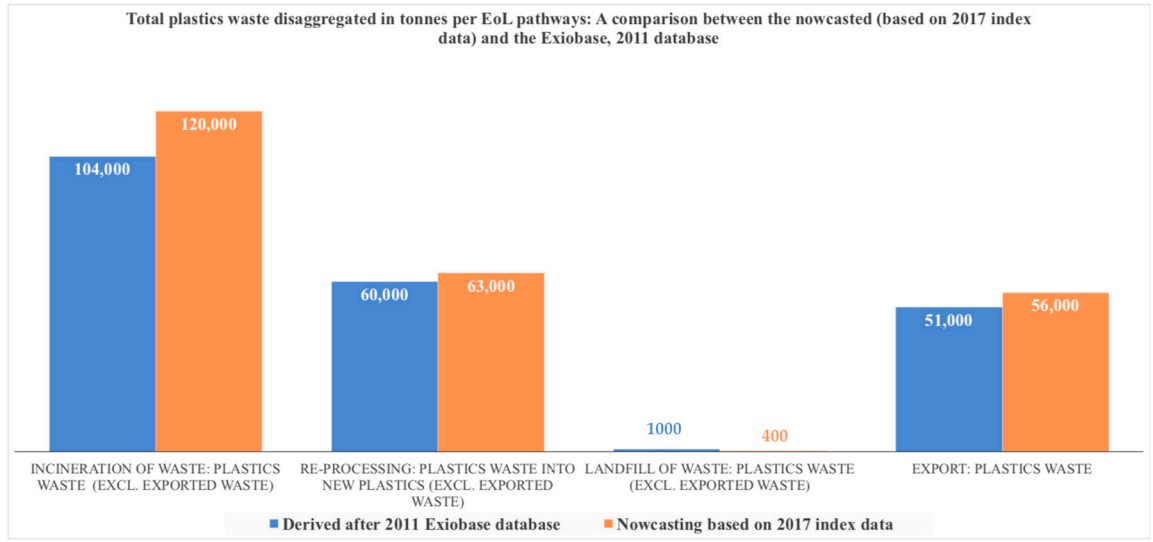

**Figure 4.** Total plastics waste disaggregated in tonnes per EoL pathway (incl. export): A comparison between nowcast results (based on 2017 index data) and the Exiobase, 2011 database.

The output from recycling for 2011 and 2017 were 54 and 57 Kt, respectively. However, assuming a ≈90% production efficiency in plastic recycling processes, we calculated that, respectively, ≈60 and 63 Kt inputs of plastics waste would be required.

Overall, incineration of plastics waste increased by ≈13% (from ≈104 Kt to 120 Kt), recycling by ≈5% (from ≈60 to 63 Kt) whereas landfilling further decreased by ≈50% (from ≈1 Kt to 0.5 Kt). See Figure 5 for the summary of the nowcast flows.

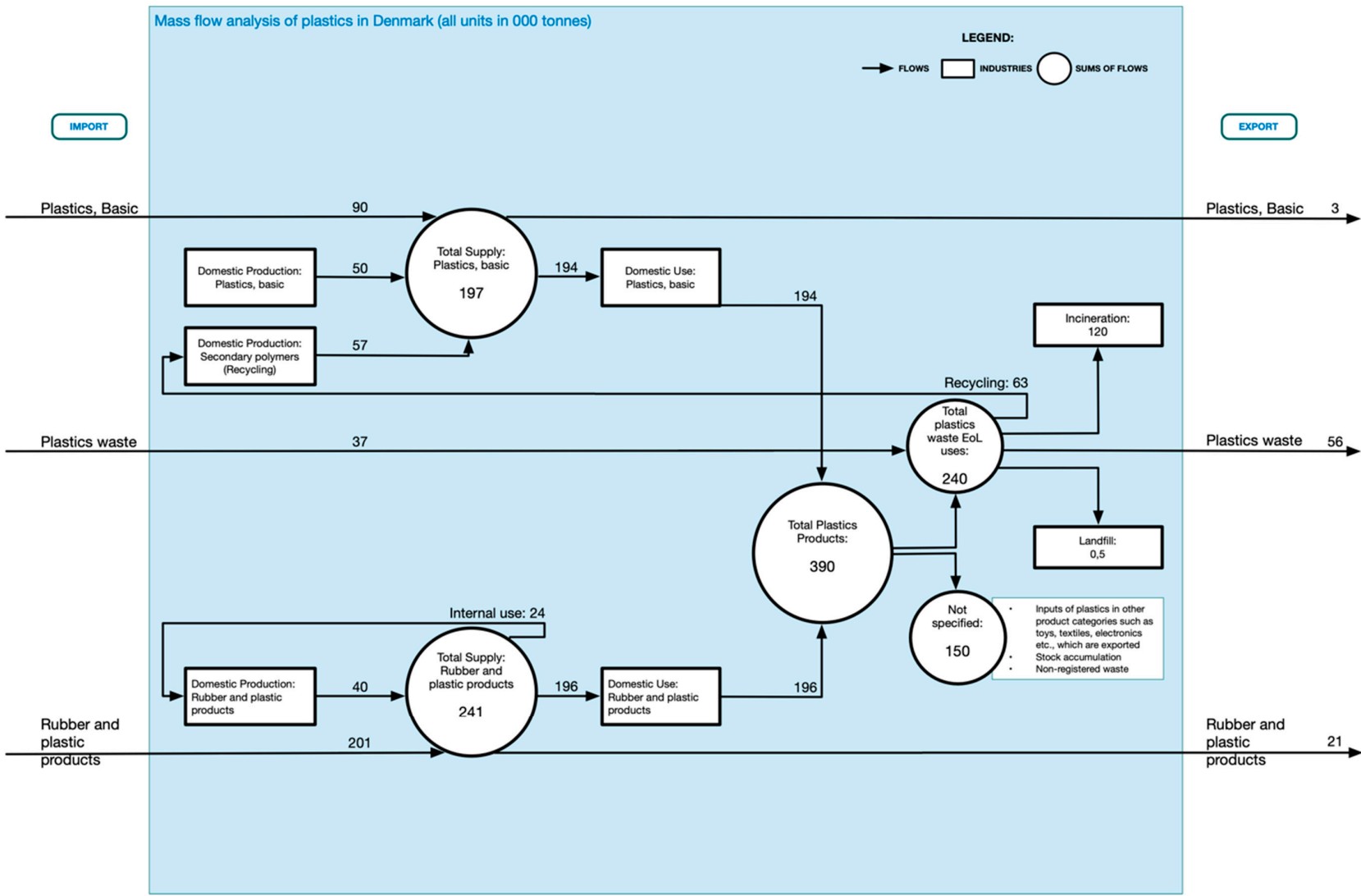

**Figure 5.** Nowcasted Danish plastics Mass Flow Analysis (MFA) based on the combination of Exiobase, 2011 data, and the 2017 trade index data.

Figure 5 shows that the total supply of plastics in Denmark by 2017 decreased by ≈20% (from ≈548–438 Kt), excl. 37 Kt imported plastics waste. Compared to the total supply for 2011, we found out that the imports of (1) plastics, basic decreased by ≈30% (39 Kt) and by ≈27% (74 Kt) that of (2) rubber and plastic products. However, domestic productions of plastics, basic had remained steady whereas, that of recycled plastics increased by ≈5% (3 Kt).

The latter also indicated a decrease of ≈22% (390 Kt) in the net use (total supply less exports) of plastics in Denmark. The domestic uses of plastics, basics, and rubber and plastic products by 2017 respectively, decreased by ≈16% (from 231–194 Kt) and 27% (from 268–196 Kt).

Furthermore, in terms of plastics waste and their EoL pathways, our results showed an increase of ≈10% (240 Kt) of plastics waste for treatment. Incineration, recycling, and export activities respectively, increased by ≈13% (from 104 to 120 Kt), 5% (60 to 63 Kt), and 9% (51–56 Kt). Landfilling decreased by ≈50% (1–0.5 Kt).

Conclusively, the difference between the net use of plastics and plastics waste for treatment showed a positive balance of ≈150 Kt. The balance is classified as ''Not specified'' however, this may account for:

- Plastics embedded in other product categories such as automobile, furniture, and construction, etc.
- Stock accumulation
- Non-registered waste.

Altogether, these account for ≈38% of the total domestic uses of plastics in Denmark.

## 5. Discussion

The results and possibilities of the plastic industry in the context of CE are discussed in this section, as well as the potential of the Exiobase as a standard database for performing MFAs.

As summarized in Figures 2 and 5, the net use of plastics in Denmark based on 2011 data was ≈499 Kt whereas, the nowcasting showed that the net use by 2017 had decreased to 390 Kt. According to the waste accounts of Exiobase, the total amount of plastics waste generated in Denmark (excl. imports of plastics waste), was ≈168 Kt. Therefore, including imports, the total supply of plastics waste in Denmark amounted to ≈235 Kt, the amount which by 2017 increased to ≈240 (incl. the 37 Kt imports of plastics waste).

Waste accounts data in Exiobase has been obtained on the basis of the total amounts of waste for incineration in Denmark combined with some relatively old and uncertain data of the composition of waste. Therefore, the proportion of plastic in the total amount of waste for incineration is associated with uncertainties.

Such uncertainties are connected to the fact that waste accounts in Exiobase are compiled based on national statistics data, which is associated with high uncertainties. These uncertainties are linked to, for instance, lack of uniformity on how waste separation is across nations, how waste data is organized and reported, as well as human errors during waste collection and some inconsistencies between the Danish and EU waste treatment categories (also see Takou et al., 2019 [20]). According to Takou et al., 2019, one of the inconsistencies associated with waste statistics is that the Danish waste treatment categories mismatch with the EU recovery and disposal codes. Another common error linked to the registration of national waste statistics is related to recycling. The use of code R1 (recycling at material level) could mean that the fraction was mistakenly sent to incineration rather than the intended recycling [20]. Additionally, as discussed earlier, plastic waste from e.g., the construction or electric and electronic equipment is respectively registered as construction waste and WEEE.

Waste accounts in Exiobase calculate the total supply of waste plus stock additions as the total use minus the sum of emissions and supply of products. The overall waste part of the supply of waste plus stock additions is determined by endogenous data on waste treatment plus trade with waste. The difference between, for example, the net use (≈499 Kt) and ≈216 Kt of the treated plastics waste by 2011 (summarized in Figure 2) or that between the nowcast 390 Kt net use and 240 Kt treated

plastics waste (in Figure 5), can alone not be explained by the stock accumulation of plastics products in the economy.

The waste quantities are most likely underestimated and hereunder classified as something else than plastics. Examples are if plastics in construction or in electrical and electronic equipment are simply classified as construction waste or electrical and electronic equipment waste (WEEE) in waste statistics; then the plastics parts hereof are not accounted as plastics waste in Exiobase. This is especially the case for multi-materials products such as radio, automobile, furniture, electrical and electronic equipment (EEE), etc. However, one solution would be to use a proxy matrix showing the material composition of these products and thereafter, extract the plastics inherent to these products. This solution has not been implemented in this study and can be considered as a topic for future research. A point of departure could be inspired by a study by [48] where they quantify (in %) the amount of plastic in a few of these products.

Another explanation relates to uncertainties associated with trade statistics data as well as cases, where plastics, basic or rubber, and plastics products have been used as feedstock for products that are exported. For example, when a pump manufacturer in Denmark uses plastic parts and then exports the pumps. In this example, the plastics parts used in the manufacturing of the pump will be registered as part of the 499 Kt herein reported as net plastic use.

Moreover, unlike the LCA convoluted waste account, plastics used in relatively long-life products will become waste more slowly. The Exiobase considers product lifetimes. An example is demonstrated in the MFA of global plastics performed by Geyer et al. (2017) [1]. In their global plastics MFA, Geyer et al. (2017) demonstrated that plastics used in long-life products will generate waste at slower rates than plastics used in short-life or single-use products such as packaging. For instance, most plastics used for packaging will enter the waste streams the same year they were produced, while plastics waste from construction originates from primary plastics produced decades earlier [1].

Other uncertainties associated with MFAs performed based on Exiobase are related to the high detailed sectoral classification as organized in Exiobase PSUT. Not all the 164 sectoral classifications represent all national economic activities included in the Exiobase. It is likely that in the Exiobase, certain activities such as value-adding processes may be classified as the production of primary forms of plastic. In this case, the value-adding activities related to the production of primary forms of plastics and basic metals were (in the Exiobase) respectively, classified as the production of plastics, basic and manufacturing of basic metals. These sectors do not exist in Denmark. We rectified this by firstly, corresponding the NACE product classification codes with the Danish codes; and secondly by visual checks, cross-referencing and confirming with the Danish Plastics Federation and the Danish industry coding system.

With Exiobase laying the basis for this MFA, we conclude that by 2011, the total supply of plastics in Denmark amounted to ≈548 Kt. Constituting the major supply, the import of rubber and plastic products and plastics, basic respectively, accounted for ≈50% and 24% whereas, the remaining 26% were from domestic productions of recycled plastics (≈10%), rubber and plastic products (≈7%) and value-adding processing of plastics, basic (≈9%).

Of the total supply, the net use of plastics in Denmark accounted for ≈91% (499 Kt) whereas, exports accounted for 5% (29 Kt). The remaining 4% accounted for internal reuse/recycling in the domestic productions of rubber and plastic products. Thus altogether, the uses of plastics amounted to ≈552 Kt. There is a 4 Kt negative balance between the total supply and uses. This may be associated with the fact that we assumed a standard 10% materials loss in recycling processes, which may vary among reprocessing companies.

Further, we nowcast our results using the trade index data to reflect the most recent (2017) flows. We found out that the total supply of plastics decreased by ≈20% (from 548–438 Kt) whereas, the major sources remained constant. The Danish imports of plastics, basic and that of rubber and plastic products decreased by ≈30% (39 Kt) and by ≈27% (74 Kt) respectively. Significant to closing the plastics

loops, we found out that domestic productions of recycled plastics output slightly increased by ≈5% (from 54–57 Kt) whereas, domestic value-adding activities of plastic, basic remained steady.

In terms of the potentials of the Danish plastic industry in the CE, calculations showed that the total supply of plastic waste by 2011 was about 216 Kt (see Figure 4). Therefore, almost half (104 Kt) is incinerated, a fourth (60 Kt) is recycled, the other fourth (51 Kt) is exported, while a tonne is landfilled. The output from the exported plastic waste remains unknown, due to their untraceable EoL pathways.

Compared to the numbers for 2017, there has been an increase of incineration of plastic waste by about 13% (16 Kt). Additionally, the export of plastic waste increased by about 9%, and a limited increase in recycling is also shown, at about 5% (see Figure 4). Thus, no significant transition has happened towards a circular economy and increased recycling of plastics. In other words, a more coordinated and collective effort to strengthen the recycling of plastics in Denmark is needed.

Further, as an EU member state, Denmark is expected to comply with the targets stipulated in the European Strategy for Plastics in a circular economy. As a medium-term strategy, the EU Commission aims to achieve 55% recycling of plastic packaging by 2030. Currently, Denmark recycles <18% of the 215 Kt of plastic packaging waste. This implies that Denmark has a demand to increase the recycling of this waste fraction by 37% (≈80 Kt) by 2030 [49]. Overall, of all its total supply of plastics waste (240 Kt), Denmark recycles 26% and the EU's strategy for plastic sets a target to quadruple the sorting and recycling capacity by 2030.

To conclude, MFA has the potential to become a foundation for policymaking and business strategy development that can point out hot spots as well as potentials for improvements. MFAs performed using Exiobase are capable of mapping resource supplies, uses, waste generation, and treatment along with the economic, environmental, and social impacts such as job creation and salary inequalities associated with these activities. However, quantitative data alone is insufficient in terms of implementation. Although an MFA gives an overview of the circular potentials of the plastic industry, other significant factors must be taken into consideration to achieve strengthened recycling.

A first step has been taken by the EU that now requires that the private and public enterprises account for the REAL recycling at the final destination to limit the leakage of the current plastic waste flows [50]. The next step is to get more focus on high-quality recycling by setting up demands to the design of plastic products in order to have fewer composite products and less heterogeneous polymers in the single product. A further step by the EU could be to give more attention to prevention and reuse strategies, and as part of this also divide in the monitoring systems between reuse and recycling instead of mixing the two levels in the waste hierarchy together.

Dialogues and collaborations between the public and private sectors, civil society organizations, institutes of research, education, and innovation are required. Therefore, MFAs data can be used both as boundary objects and a point of departure where significant hotspots and relevant stakeholders and actors for collaboration on circular plastics are identified. Other factors include refusal to use dispensable plastic products, reduction of plastic consumption, and increased use of homogenous polymer types per product during the design phase. Additionally, there is a need for municipalities to align their waste sorting, registration, and infrastructure systems.

Methodologically, compared to other methods based on either solely statistical or mixed sources of data, (see introduction), Exiobase has the potential to qualify as a standard method for performing MFAs. Exiobase is built upon consistent monetary and physical SUTs thereby, enabling the coupling of economic data with physical MFA data and energy data. These elements make it possible to maintain the mass balance criterion and is methodologically reproducible. Additionally, Exiobase also presents economic and environmental data in a format that is consistent with the recommended accounting systems proposed by the United Nations (UN) System of Environmental-Economic Accounting (SEEA) [33].

Given the current production, uses and waste generation of plastics, Denmark has a significant potential for implementing the CE strategies, in this case, recycling. The final consumption (from

household) represent great challenges and potentials. Households in Denmark consume 75% of recyclable plastics such as low-density polyethylene (LDPE) and high-density polyethylene (HDPE).

Final consumption, as well as other sectors overall, environmental improvements can be achieved in various ways. This part of the discussion focuses on the potential for improvement. Greenhouse gas emissions were chosen to be focused on as a relevant indicator of the environmental impact of plastics. Other indicators could also be relevant, however, greenhouse gas emissions are considered to be the most important indicator, where the uncertainties in the underlying LCA data and models are least.

Listed below are various typical strategies for how environmental improvements can be achieved within plastic areas:

- Plastic is replaced with less environmentally harmful materials
- The consumption of plastic is reduced through resource efficiency, eg avoiding unnecessary use of packaging
- The service life/degree of use of plastic products is increased, so that the net consumption is thereby reduced, e.g., when plastic bags are used several times
- Recycling of plastics is increasing

When the above strategies are evaluated in relation to how much they can contribute to reducing the environmental impact, it is important to also include rebound effects. Most of the above strategies will result in a reduction in costs associated with the service that the plastics fulfill. This will, compared to other alternatives, mean that there are more financial benefits for other consumption, which may be associated with significant environmental impacts. However, rebound effects are not assessed in this report, but it is emphasized that they are potentially important for the outcome of how large reduction potentials and different strategies can have.

In terms of mapping the environmental impacts and potentials for the CE plastics in Denmark (focused on recycling), Exiobase contains three different types of treatment for plastic waste i.e., recycling, incineration, and landfill. Below are the average GHG emissions per kg of treated plastic waste (GWP100 cf. IPCC's 'Fifth Assessment Report' from 2013) [51,52].

- Recycling in Denmark: $-2.14$ kg $CO_2$ equivalents/kg plastic waste
- Incineration in Denmark: $0.82$ kg $CO_2$ equivalents/kg plastic waste
- Landfill in Denmark: $0.32$ kg $CO_2$ equivalents/kg plastic waste

For example, the production of 1 kg of new average plastic for the Danish market emits 3.82 kg of $CO_2$ equivalents. It appears from the above strategies that recycling can significantly reduce greenhouse gas emissions from production and disposal. For example, overall, the production and subsequent combustion of plastics emit $3.82 + 0.82 = 4.64$ kg $CO_2$ equivalents, while production and subsequent recycling only emit $3.82 - 2.14 = 1.68$ kg $CO_2$ equivalents.

The advantage of recycling is that it displaces the production of new plastic, which emits far more $CO_2$ eq. than it requires to collect, clean, and process plastic waste into new plastic. Displacement also occurs when burning plastic waste, but the greenhouse gas emissions from the displaced electricity and heat are far less than the emissions from the incineration process, which emit about 2.5 kg $CO_2$ per kg plastic waste. Landfilling emits fewer greenhouse gas emissions than incineration because the carbon in the plastic is not incinerated to form $CO_2$.

Attention is drawn to the fact that the stated advantages of recycling above cover large differences in what goes on in practice. First, the collection of plastic waste does not necessarily mean that plastic is recycled. This is the case when different types of plastic cannot be disassembled, and recycling into new plastic is thus either not feasible or it involves heavy downcycling.

If recycling is impossible, then the plastic will be incinerated and both costs and emissions from collection and sorting may be higher than incineration. Downcycling means that the recycled plastic cannot be recycled for the same type of purpose as the original products. Downcycling is not necessarily bad, as long as the displaced products caused by recycling have the same marginal

impact as the manufacture of new plastics for new products. For example, recycling processed plastics with impurities for garden furniture will displace plastics from the same market as products with requirements for clean plastics. Although there has been downcycling from pure to impure plastic, the same market is generally affected. However, this type of downcycling is only a good idea as long as the market for unclean plastic for plastic products is not saturated.

Downcycling can mean that recycling plastic displaces completely different materials. For example, recycling unclean plastic for park benches will typically displace wood, while recycling car tires for playgrounds will typically displace sand.

The above conditions regarding downcycling are not included in the stated −2.14 kg $CO_2$ equivalents per kg of plastic waste. Therefore, the actual potential may be less.

It should also be noted that different types of plastics are associated with different emissions. For example, polystyrene emits about 75% kg more $CO_2$ than polyethylene, and nylon emits about four times as much as polyethylene.

## 6. Conclusions

EE MRIO database is a tool for indicating the resource extraction and uses, the product uses, and their associated environmental impacts. This then can provide a data foundation for making business strategies and for policymaking decision processes for both the public and private sectors. Exiobase was chosen for this research due to its scientific robustness, high sectoral details, and coverage of the influential economic activities.

Exiobase is a database for performing scientifically consistent material flow analyses. However, even its latest version, the Exiobase is compiled on HSUTs based on 2011 data. Therefore, there is still a need to update the database to reflect the current economic activities. Note that the use of trade index data may be associated with some uncertainties as mentioned in the methodology section. Nonetheless, data on the major sources, total supply, and uses of plastics in Exiobase consistently reflect on sectoral economic activities.

Furthermore, waste account algorithms in Exiobase consider the lifetime of products. As presented in Section 2, Exiobase algorithms take an unconvoluted approach that differs from, for example, the convoluted LCA and mainstream methods to waste accounts. The Exiobase takes into account products' lifetime and depreciation values. Hence, plastics used in long-life products, such as automobile and construction, within the Exiobase are not accounted for as waste in the very same year they enter the use phase. All these above points highlight that the Exiobase remains the concise method for performing scientifically justifiable mass flow analyses.

This research recommends and supports a unified system [48] of plastic composition in multi-material products such as electronics and automobiles, waste categorization, organization, and data collection. This study hinted at the necessity to consider the proportion of resources used in all products and documented illegal trade of waste. Such discrepancies and errors are confirmed through the law of mass balance upon which the Exiobase is constructed. Additionally, this research highly recommends that resources be channeled towards continuously updating the Exiobase because statistical trade index data increases the uncertainties of the results. Considering the discussion and conclusion of this paper, Exiobase is a scientifically robust EE MRIO database for performing mass flow analyses.

**Author Contributions:** Conceptualization, E.V. and A.R.; methodology, E.V.; software, E.V.; validation, E.V.; formal analysis, E.V., E.T.and A.R.; investigation, E.V. and E.T.; resources, E.V., E.T. and A.R.; data curation, E.V. and E.T.; writing—original draft preparation, E.V.; writing—review and editing, E.V., E.T. and A.R.; supervision, A.R.; revision and project administration, E.V. All authors have read and agreed to the published version of the manuscript.

**Funding:** This research received no external funding.

**Conflicts of Interest:** The authors declare no conflict of interest.

## Appendix A

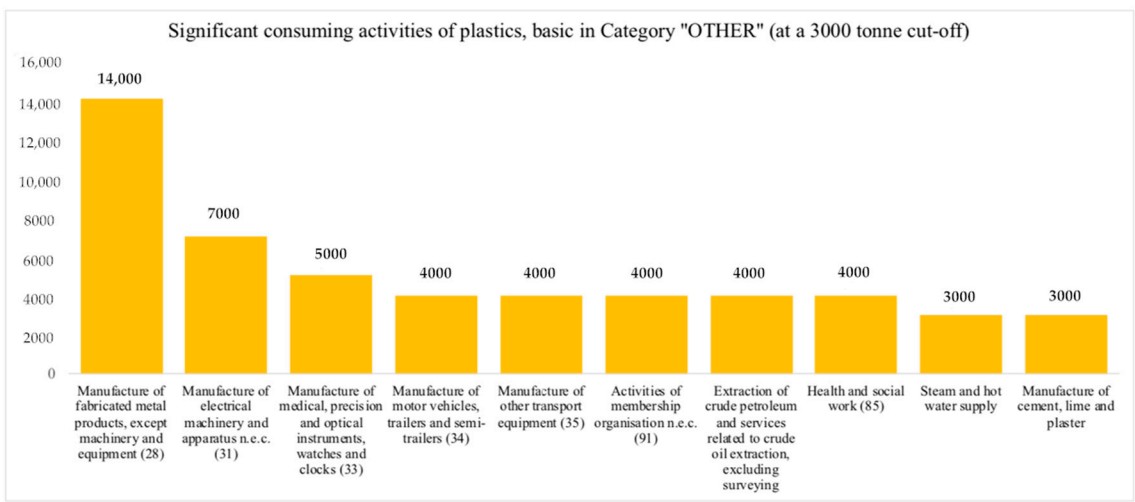

**Figure A1.** The most significant consuming activities of plastics, basic in category ''OTHER'' (at a 3 Kt cut-off). Based on the 2011 database.

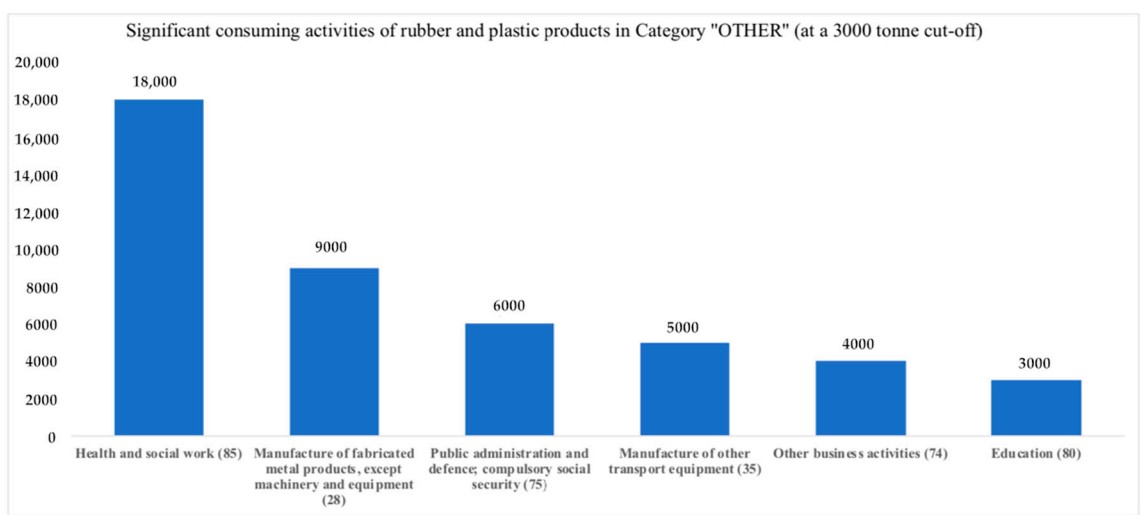

**Figure A2.** The most significant consuming activities of rubber and plastic products in category OTHER (at a 3 Kt tonne cut-off). Derived after 2011 database.

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
