# Peer review of "Danish Plastic Mass Flows Analysis"

_sustainability, doi:10.3390/su12229639_

Round 1

Reviewer 1 Report

This is a very interesting study that brings forth relevant quantitative and qualitative data and provides relevant analyses and visualization options.

Some improvement suggestions:

  • figures 1 to 4 appear empty, they should be fixed
  • text on page 13 should be integrated with the text before, to provide a better placement for figure 5
  • the title of the manuscript and of the form should be the same
  • if CE will be featured in the title, the paper should include estimations of the impact of CE approaches on the MFA

Author Response

Many thanks for for your very useful reviews. The manuscript is now edited as requested. 

Reviewer 2 Report

original study and full of reflex spikes, tackles a current issue with a known but well structured and explained methodology. important results; complete and current bibliography; exemplary graphics

Author Response

Many thanks for your very useful reviews. Please, find attached a new version revised after your reviews.
